# Episodic Future Thinking about Smoking-Related Illness: A Preliminary Investigation of Effects on Delay Discounting, Cigarette Craving, and Cigarette Demand

**DOI:** 10.3390/ijerph19127136

**Published:** 2022-06-10

**Authors:** Perisa Ruhi-Williams, Mary J. King, Jeffrey S. Stein, Warren K. Bickel

**Affiliations:** 1Fralin Biomedical Research Institute, 1 Riverside Circle, Roanoke, VA 24016, USA; perisaruhi@gmail.com (P.R.-W.); kingm20@vtc.vt.edu (M.J.K.); wkbickel@vtc.vt.edu (W.K.B.); 2Virginia Tech Carilion School of Medicine, 4 Riverside Circle, Roanoke, VA 24016, USA; 3Irvine Medical Center, University of California, Orange, CA 92697, USA; 4Graduate Program in Translational Biology, Medicine and Health, Virginia Tech, Blacksburg, VA 24016, USA

**Keywords:** cigarettes, delay discounting, episodic future thinking, smoking-related illness

## Abstract

Cigarette smokers show excessive delay discounting (devaluation of delayed rewards), which may contribute to tobacco use disorder. Episodic future thinking (EFT), or mental simulation of future events, has been shown to reduce both delay discounting and laboratory smoking behavior. Traditionally, EFT involves vividly imagining positive future events. In this preliminary investigation, we examined the effects of EFT specifically about smoking-related illness (SRI) on delay discounting, cigarette craving, and behavioral economic demand for cigarettes. In a 2 (episodic thinking) × 2 (smoking-related illness) factorial design, we randomly assigned smokers from Amazon Mechanical Turk to one of two EFT groups: EFT alone or EFT + SRI; or one of two episodic “recent” thinking (ERT) control groups: ERT alone or ERT + SRI. Both EFT groups generated and imagined positive future events, while both ERT groups imagined real events from the recent past. Both EFT + SRI and ERT + SRI groups imagined these events while also experiencing SRI symptoms. Participants then completed assessments of delay discounting, cigarette craving, and measures of cigarette demand. We observed significant main effects on delay discounting of both EFT (reduced discounting) and SRI (increased discounting), as well as significant main effects of both EFT and SRI on cigarette craving (in both cases, reduced craving). No significant main effect of EFT was observed on cigarette demand measures, although we observed a main effect of SRI on quantity of demand when cigarettes were free (Q0) (reduced demand). In all analyses, we observed no significant EFT × SRT interactions, indicating that these variables operate independently of one another. These methods may be adapted for use in clinical treatment to aid in smoking cessation interventions.

## 1. Introduction

Cigarette smoking is the leading preventable cause of mortality. Quitting smoking reduces the risk of developing and dying from smoking-related illnesses [1]. For example, cigarette smoking is the greatest risk factor for the development of lung cancer, which is the first and second leading cause of cancer deaths worldwide in men and women, respectively [2]. Additional research is needed to gain a more thorough understanding of the factors that influence the success in smoking cessation.

Delay discounting refers to the tendency for people to devalue delayed reinforcement [3], and it provides a measure of individuals’ preference for larger, delayed outcomes over smaller, immediate outcomes. Excessive delay discounting is a behavioral marker of cigarette smoking and other forms of addiction [4]. For example, in cross-sectional studies, cigarette smokers show elevated rates of delay discounting [5,6]—a finding replicated in opioid users [7], alcohol-dependent participants [8], and problem gamblers [9]. Likewise, in prospective studies, excessive delay discounting in early adolescence predicts the initiation of cigarette smoking in late adolescence [10].

Based on these cross-sectional and prospective findings, some have suggested that excessive delay discounting may play a causal role in initiating and maintaining cigarette smoking and other addictive behavior [11]. Specifically, the rapid devaluation of the negative, delayed consequences of smoking (e.g., lung cancer) may increase the valuation of the immediately rewarding effects of nicotine. Further evidence for this hypothesis comes from experimental studies in which variables that reduce delay discounting also reduce the addictive behavior with which discounting is correlated. For example, episodic future thinking (EFT) is a type of prospection that involves mental simulation of future events [12]. In laboratory studies, EFT reduces both delay discounting [13,14,15,16,17] as well as caloric intake in individuals with overweight and obesity [13] and economic valuation of alcohol in individuals with alcohol use disorder [15]. Most relevant to the present study, EFT has been shown to reduce the valuation of cigarettes and laboratory-based cigarette self-administration [16,18].

These experimental findings further implicate a causal role of delay discounting in cigarette smoking and suggest that laboratory-based EFT methods may be adapted for use in clinical settings to aid in smoking cessation. For example, to supplement existing treatments for smoking cessation (e.g., cognitive behavioral therapy), individuals may be prompted to engage in EFT in the natural environment via ecological momentary intervention [19,20] during times of day they are most vulnerable to cigarette cravings. However, a more thorough understanding of EFT’s effects on smoking is needed before EFT can be used effectively in clinical settings. Toward that end, we aimed in the present study to examine a method to make EFT more effective in reducing the motivation to smoke. Specifically, most studies on EFT have examined the effects of positive future events on delay discounting and other measures. However, if excessive delay discounting plays a causal role in cigarette smoking (as outlined above), a fundamental problem is that the long-term and the inherently negative effects of smoking are too delayed to discourage smoking in the present. Thus, engaging in a form of EFT in which the individual pre-experiences the effects of smoking-related illness (SRI) may render those negative outcomes more vivid, thus allowing delayed outcomes to more effectively guide motivation to smoke.

Thus, in the present study, we used an online sample to examine the effects of EFT involving SRI (specifically lung cancer) on delay discounting and two different measures of motivation to smoke: cigarette craving and behavioral economic demand for cigarettes (i.e., consumption as a function of price; a measure of cigarette valuation). We also compared these effects to EFT without SRI, as used commonly in the literature. We compared both EFT conditions to two episodic *recent* thinking (ERT) control conditions (with and without SRI) in which participants imagined real events that occurred over the past several days [21,22]. We hypothesized that, consistent with prior data, we would replicate prior effects of EFT on delay discounting compared to ERT [16] and extend these findings to show that EFT also reduces cigarette craving and demand. We also hypothesized that adding SRI symptoms (i.e., vividly imagining a future event while also experiencing lung cancer symptoms) would further reduce cigarette craving and demand. We had no specific hypotheses, however, regarding possible interactions between the EFT and the SRI conditions.

## 2. Methods

### 2.1. Participants

Participants (*N* = 199) were recruited through Amazon Mechanical Turk (AMT). This crowdsourcing website allows human workers to complete posted Human Intelligence Tasks (HITs) for compensation. Participants initially completed a three-item screening questionnaire to determine eligibility. In order to be eligible for this study, participants had to report that they: (1) were a current cigarette smoker, (2) smoked ≥ 5 cigarettes per day, and (3) had not previously been diagnosed with an SRI. Additional inclusion criteria, determined automatically through AMT, required that participants (4) had a 90% acceptance rate on previous HITs indicating that their work has been of sufficient value to be accepted by requesters at least 90% of the time, (5) resided in the United States, and (6) were at least 18 years of age. In addition, participants were excluded from eligibility if they had previously been diagnosed with an SRI. Participants were compensated $2.00 for completion of the task, with an additional $4 bonus for meeting standardized diagnostic criteria for behavioral economic demand (see below).

After screening, participants were randomly assigned to one of four groups: EFT (*n* = 50), ERT (*n* = 51), EFT + SRI (*n* = 50), or ERT + SRI (*n* = 48). Table 1 provides the demographic and smoking characteristics of participants assigned to each group. We observed no significant differences between groups in any measure.

The sample size was chosen to approximate similar group sizes in prior studies of EFT [6]. In a sensitivity power analyses, approximately 200 participants provided 95% power to detect a medium effect size in analysis of variance, assuming four groups and alpha = 0.05.

**Table 1 ijerph-19-07136-t001:** Participant characteristics.

	Group				
Characteristic	EFT, *n* = 50	EFT-SRI, *n* = 50	ERT, *n* = 51	ERT-SRI, *n* = 48	Overall, *n* = 199
**Gender** *n* (%)					
Male	29 (58%)	27 (54%)	25 (49%)	25 (52%)	106 (53%)
Female	21 (42%)	23 (46%)	26 (51%)	23 (48%)	93 (47%)
**Race***n* (%)					
White/Caucasian	40 (80%)	46 (92%)	45 (88%)	44 (92%)	175 (88%)
Black/African American	3 (6.0%)	2 (4.0%)	4 (7.8%)	2 (4.2%)	11 (5.5%)
Asian	5 (10%)	1 (2.0%)	1 (2.0%)	1 (2.1%)	8 (4.0%)
Other/Did not specify	2 (4.0%)	1 (2.0%)	1 (2.0%)	1 (2.1%)	5 (2.5%)
**Ethnicity** *n* (%)					
Not Hispanic/Latino	47 (94%)	44 (88%)	46 (90%)	44 (92%)	181 (91%)
Hispanic/Latino	3 (6.0%)	6 (12%)	5 (9.8%)	4 (8.3%)	18 (9.0%)
**Education** *n* (%)					
High school or less	18 (36%)	18 (36%)	20 (39%)	20 (42%)	76 (38%)
Associate’s degree	5 (10%)	6 (12%)	7 (14%)	4 (8.3%)	22 (11%)
Bachelor’s degree	22 (44%)	21 (42%)	19 (37%)	20 (42%)	82 (41%)
Post-graduate degree	5 (10%)	5 (10%)	5 (9.8%)	4 (8.3%)	19 (9.5%)
**Mean Household income** (±SD)	71,300 ± 41,303	52,900 ± 35,255	62,451 ± 38,799	47,708 ± 32,255	58,719 ± 37,926
**Mean Age** (±SD)	34.98 ± 8.71	36.46 ± 10.65	34.82 ± 10.35	33.27 ± 9.49	34.90± 9.82
**Mean cigarettes/day** (±SD)	12.62 ± 6.42	13.70 ± 6.87	14.00 ± 5.70	14.31 ± 7.57	13.65 ± 6.64
**Mean FTND score** (±SD)	4.20 ± 2.14	4.20 ± 2.18	4.59 ± 2.29	4.62 ± 2.39	4.40 ± 2.24

### 2.2. Procedures

Participants first completed an initial demographic and smoking history questionnaire. Next, participants completed a task designed to generate vivid EFT or ERT events (either with or without SRI symptoms) and related text cues. Following these generation tasks, participants completed a delay-discounting task, a cigarette craving questionnaire, and a cigarette purchase task (order randomized). Following these tasks, participants completed an affect scale. The entire study, including initial questionnaire and task completion, took 35–45 min to complete (an effective rate of $8–$10/h).

### 2.3. EFT Generation Task

Participants in the EFT group used a self-guided generation task [17] to elicit positive events that they were looking forward to at three different time points in the future (1 month, 6 months, 1 year). They were asked to provide detailed text descriptions of these future events including who would be with them, where they would be, and how they would be feeling. Participants were shown examples of “good” cues that were set in the future, positive, and detailed, as well as examples of “bad” cues that were vague and lacked episodic details. After typing their cue for each time point, participants were asked to rate their cue on a scale of 1 (very low) to 5 (very high) on vividness, positivity, and importance dimensions.

### 2.4. ERT Generation Task

Participants in the ERT group followed the same procedures for cue generation as the EFT group, except that they were asked to imagine positive events in the recent past at three different time points (1 day, 6 days, 12 days). The ERT condition, used frequently in studies of EFT and delay discounting [21,22] serves to isolate the effects of prospection in active EFT by ensuring that episodic content in both groups engages episodic memory, features personalized detail, and is hence matched for vividness.

### 2.5. EFT + SRI and ERT + SRI Generation Tasks

Participants in the EFT + SRI and ERT + SRI groups followed the same procedures for cue generation as the EFT and ERT groups, respectively. After completing the initial cue generation, participants were informed that a primary cause of lung cancer is cigarette smoking and were asked to add one or more vivid symptoms of smoking-related illness (SRI) to their positive events. Participants were provided a list of symptoms of lung cancer as follows: difficulty breathing, feeling extremely tired or weak, coughing up blood, chest pain, coughing up phlegm or mucus, and harsh sounds with each breath. Examples of cues with added SRI symptoms were also shown.

### 2.6. Delay Discounting Task

Following the cue generation task, participants were presented with a set of hypothetical choices between smaller, immediate rewards and larger, delayed rewards. During each decision, participants were instructed to read and think about their self-generated events. Each choice was displayed with the self-generated event that corresponded approximately to the delay for that choice. For example, when choosing between $50 now or $100 in 1 year, EFT and EFT + SRI participants were instructed to think about their future events that will occur in about 1 year (the most distal future time point). In contrast, at this delay, the ERT and ERT + SRI groups thought about the event occurring 12 days ago (the most distal recent time point).

Depending on the participant’s preceding choice, the amount of the smaller reward was either increased or decreased on the next trial [23], until an indifference point was reached. At this indifference point, which reveals the discounted value of the larger, delayed reward, the subjective value of both rewards is approximately equal to the participant. For example, indifference between $50 now and $100 in 1 year reveals that the delay has caused the larger reward to lose half of its subjective value. This titration process was repeated at five delays (1 day, 1 week, 1 month, 6 months, and 1 year), in random order.

Two quality control questions immediately followed the delay discounting task (in random order), similar to those used previously [24,25,26]. Using identical question text and formatting as the delay discounting questions, one of these questions asked participants to choose between $100 in 1 day and $0 now; the other question asked participants to choose between $0 in 1 day and $50 now. Choice for the “now” option in either question was interpreted as inattention.

### 2.7. Cigarette Craving Questionnaire

Participants completed the Questionnaire of Smoking Urges-Brief (QSU) [27] to measure cigarette craving. The QSU is a 10-item questionnaire which asks participants to rate their agreement with a number of statements about smoking (e.g., “All I want right now is a cigarette.”) using a 7-point Likert scale. Higher scores reflect greater cigarette craving. As in the delay-discounting task, participants read and thought about their self-generated events while answering the questionnaire. Consistent with prior work [18], only the one-year cue was used.

### 2.8. Cigarette Purchase Task

Participants completed a cigarette purchase task [28,29] to estimate behavioral economic demand for cigarettes. Here, participants reported the number of cigarettes they would like to purchase across a range of 13 prices (starting at $0, and incrementing from $0.03–$64.00 per cigarette, in approximately log_2_ intervals). Participants were asked to assume that cigarettes were for use during a single 24 h period, could not be shared or stockpiled, and could not be accessed from any other source. As in previous tasks, participants read and thought about their self-generated events while answering each question (one-year cue only).

### 2.9. Positive and Negative Affect Scale

Finally, participants completed the Positive and Negative Affect Scale (PANAS) [30], while reading and considering their events (one-year cue only). The PANAS is a 20-item questionnaire in which participants use a five-point Likert scale to rate how a number of different emotional words (e.g., “afraid” or “excited”) describe their affective state.

### 2.10. Data Analysis

Demographic and smoking measures were compared between groups using either logistic regression (dichotomous data) or two-way ANOVA (continuous data).

*Delay discounting.* Delay discounting data were first subjected to a preliminary analysis using standardized diagnostic criteria [31] to detect the presence of data that were not systematically affected by delay. These criteria assume only that systematic data show: (1) a global reduction in discounted value of at least 10% at the longest delay (i.e., the *trend* criterion) and (2) consistent local effects of contiguous delays, with no or few increments in delay containing an increase in discounted value (i.e., the *bounce* criterion). Although these criteria are often used as a basis for data exclusion, they were used primarily for descriptive purposes (i.e., not data exclusion) in the present study, as prior research suggests that the presence of data that are not systematically influenced by delay may be a direct effect of EFT [16]. That is, violations of the trend criterion (i.e., no global reduction in discounted value) could be an expected outcome of an intervention designed to reduce delay discounting, particularly when relatively small values of the longest delay are used (e.g., one year). Likewise, violations of the bounce criterion (i.e., inconsistent local effects of contiguous delays) may be influenced by heterogeneity in the efficacy of cues across delays, which may cause intermittent bounce even when participants understand task instructions and are paying attention. The frequency of nonsystematic data was compared between groups using logistic regression.

Beyond these criteria, one participant failed one of the two quality control questions presented after the delay-discounting task, indicating inattention. Data from this participant were excluded from all delay-discounting analyses.

To estimate delay discounting, we calculated the area under participants’ delay discounting curves (area under the curve or AUC) expressed as a proportion of the maximum possible area [32]. This measure may range from 0 (maximum discounting) to 1 (no discounting).

*Cigarette craving.* To estimate cigarette craving, we summed participants’ agreement ratings on the 10 items from the QSU. This measure may vary from 10 (minimum craving) to 70 (maximum craving).

*Cigarette demand.* Cigarette demand data were first subjected to preliminary analysis using standardized criteria [33] to detect the presence of data that were not systematically affected by price. These criteria, similar to those used for delay discounting, assume that systematic data show: (1) a global reduction in purchasing from the first price to the last, with the size of this reduction proportional to the range of prices examined (*trend* criterion), and (2) consistent local effects of contiguous prices on purchasing, defined by both the *bounce* criterion (as above) and the *reversal from zero* criterion (i.e., no instance of zero purchasing is followed by non-zero purchasing at higher prices). All data passed the bounce and the reversal from zero criteria. In contrast, 17 participants’ demand curves violated the trend criterion. Further inspection revealed this was, in all cases, due to zero purchasing at all prices. These “null demand” data could be the logical consequence of behavioral interventions designed to reduce demand, unlike more typical violations of the trend criterion (e.g., increasing demand as a function of price) [33]. Thus, all null demand data were retained in analyses, although the frequency of null demand data was compared between groups using logistic regression.

Beyond these criteria, one participant in the ERT-SRI group indicated that they would consume >19,999 cigarettes in 24 h at each of the first five prices. Data from this participant were excluded from all demand analyses.

Due to the presence of null demand data, which cannot be fitted using traditional exponential behavioral economic models [34,35], demand measures were estimated exclusively using observed values of Q_0_ (quantity of demand at $0, unconstrained by price), OMax (maximum expenditure), and PMax (price at which maximum expenditure occurs). In all cases, higher values of these measures reflect greater demand for cigarettes. Together, these measures represent both factors comprising the latent structure of cigarette demand [36], with PMax reflecting the persistence of demand (i.e., price sensitivity), Q_0_ reflecting amplitude of demand, and OMax reflecting aspects of both persistence and amplitude. All measures were nonnormally distributed (positive skew) and were thus natural log-transformed before analysis. A constant of 0.5 was added to all values to accommodate log transformation of 0 values.

*Affect.* Finally, to estimate affect, scores (1 to 5) on individual items from the PANAS were summed within positive and negative affect subscales, yielding separate aggregate scores for positive and negative affect (possible range 10 to 50 for each subscale). To yield a single value describing the balance of positive to negative affect, we subtracted negative affect from positive affect scores to yield a difference score (possible range: −40 to 40). These difference scores served as our dependent measure, with scores below 0 reflecting greater negative affect relative to positive affect and scores above 0 reflecting the opposite. We used a simple difference score because this measure was normally distributed, whereas ratio or proportion measures of positive and negative affect used previously [37,38] were positively skewed or bimodal.

*Analysis.* For each of the measures listed above, we analyzed the effects of EFT and SRI using separate 2 (EFT vs. ERT) × 2 (SRI vs. no SRI) analyses of covariance (ANCOVA), including all possible main effects and interactions. Partial eta-squared effect sizes were also calculated, with values of η_p_^2^ = 0.01, 0.06, and 0.14 reflecting small, medium, and large effects, respectively [39].

## 3. Results

### 3.1. Demographic Characteristics

Table 1 shows the demographic and the smoking characteristics of participants assigned to each group. We observed no differences between groups in any measure, except for household income in which significantly higher incomes were observed in participants assigned to the SRI conditions (i.e., a main effect of SRI; *F*(1, 195) = 9.911, *p* = 0.002; η_p_^2^ = 0.048); this difference was likely due to chance because group assignment was randomized. No significant main effect of EFT nor EFT × SRI interaction was observed (in both cases, *p* > 0.180). Income was included as a covariate in ANCOVA (Appendix A).

### 3.2. Delay Discounting

Figure 1A depicts the main effects of both EFT and SRI on delay discounting. We observed significant main effects of both episodic thinking (*F*(1, 193) = 8.203, *p* = 0.005; η_p_^2^ = 0.041) and SRI (*F*(1, 193) = 6.109, *p* = 0.014; η_p_^2^ = 0.032) on AUC, with these variables affecting discounting in opposite directions. That is, EFT (regardless of SRI condition) decreased discounting and SRI symptoms (regardless of EFT condition) increased discounting. We did not observe a significant EFT × SRI interaction (*F*(1, 193) = 0.940, *p* = 0.333; η_p_^2^ = 0.005), indicating that these variables influenced delay discounting independently of one another. The covariate effect of income was not significant (*F*(1, 193) = 0.002, *p* = 0.966; η_p_^2^ = 0.000).

Table 2 provides the frequency of nonsystematic delay discounting data for each of the criteria. Although violations of the trend criterion were nominally more frequent in the two EFT groups compared to the two ERT groups, these violations were not significantly associated with an EFT condition (OR = 2.694, 95% CI [0.990, 7.330], *p* = 0.052) or an SRI condition (OR =0.571, 95% CI [0.156, 2.092], *p* = 0.398), or the EFT × SRI interaction (OR = 0.454, 95% CI [0.083, 2.493], *p* = 0.454). Likewise, violations of the bounce criterion were not significantly associated with an EFT condition, an SRI condition, or the EFT × SRI interaction.

### 3.3. Cigarette Craving

Figure 1B depicts the effects of both EFT and SRI on cigarette craving. We observed a significant main effect of EFT (*F*(1, 194) = 5.105, *p* = 0.025; η_p_^2^ = 0.026) and SRI (*F*(1, 194) = 5.932, *p* = 0.016; η_p_^2^ = 0.030) on craving scores. These effects were unidirectional, indicating that both EFT and SRI symptoms reduced craving (regardless of the opposing conditions). Again, we did not observe a significant EFT × SRI interaction (*F*(1, 194) = 0.228, *p* = 0.633; η_p_^2^ = 0.001). Finally, the covariate effect of income was not significant (*F*(1, 193) = 0.911, *p* = 0.341; η_p_^2^ = 0.005).

**Figure 1 ijerph-19-07136-f001:**
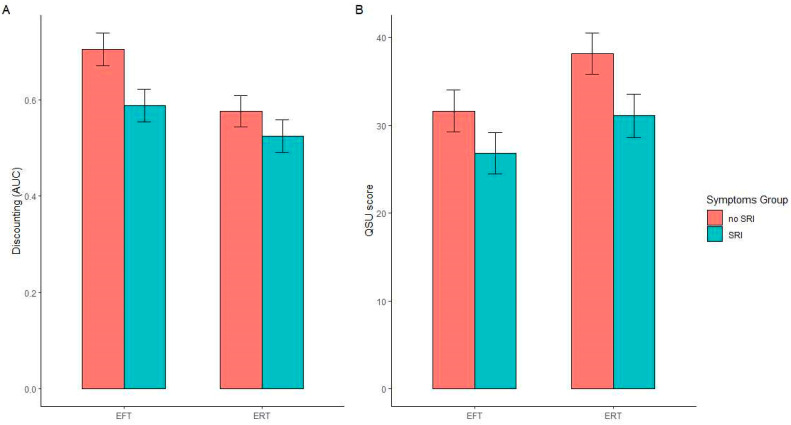
Mean covariate-adjusted values of delay discounting area under the curve (AUC; panel (**A**)) and cigarette craving in the Questionnaire on Smoking Urges-Brief (QSU; (panel (**B**)) in episodic future thinking (EFT) and smoking-related illness (SRI) groups. Higher values of AUC reflect less discounting of the delayed reward. Higher values of QSU score reflect greater cigarette craving. Error bars represent standard error of the mean. Significant main effects of EFT and SRI were observed on both measures (in both cases, *p* < 0.050), with no significant EFT × SRT interactions (in both cases, *p* > 0.330).

### 3.4. Cigarette Demand

Figure 2 depicts EFT and SRI’s effects on demand measures, including Q_0_, PMax, and OMax (panels A, B, and C, respectively). We did not observe significant main effects of EFT on any of the demand measures: Q_0_, or quantity of demand at $0 (*F*(1, 193) = 2.707, *p* = 0.102; η_p_^2^ = 0.014); OMax, or maximum expenditure (*F*(1, 193) = 2.082; *p* = 0.151; η_p_^2^ = 0.011), and PMax, or price at maximum expenditure (*F*(1, 193) = 0.033; *p* = 0.856; η_p_^2^ = 0.000). However, we did observe significant main effects of SRI on Q_0_ (*F*(1, 193) = 4.808, *p* = 0.03; η_p_^2^ = 0.024), with lower Q0 values observed in the SRI compared to no SRI groups. However, no significant main effects were observed on either OMax (*F*(1, 193) = 1.731, *p* = 0.190; η_p_^2^ = 0.009) or PMax (*F*(1, 193) = 0.095, *p* = 0.758; η_p_^2^ = 0.000). No significant EFT × SRI interactions were observed in any measure (in all cases, *F* < 0.379, *p* > 0.540). Likewise, the covariate effect of income was nonsignificant in all demand analyses (in all cases, *F* < 1.733, *p* > 0.190).

The direction of the significant main effect of SRI on Q_0_ indicated that SRI symptoms (regardless of EFT condition) reduced the level of cigarette demand when free.

Table 2 provides the frequency of nonsystematic demand data for each of the criteria. Although violations of the trend criterion (in all cases, null demand) were nominally more frequent in the two SRI groups compared to the two groups without, these violations were not significantly associated with an SRI condition (OR = 0.566 (0.919, 22.727), *p* = 0.063), an EFT condition (OR = 1.161 [0.371, 3.356], *p* = 0.896), or the EFT × SRI interaction.

**Figure 2 ijerph-19-07136-f002:**
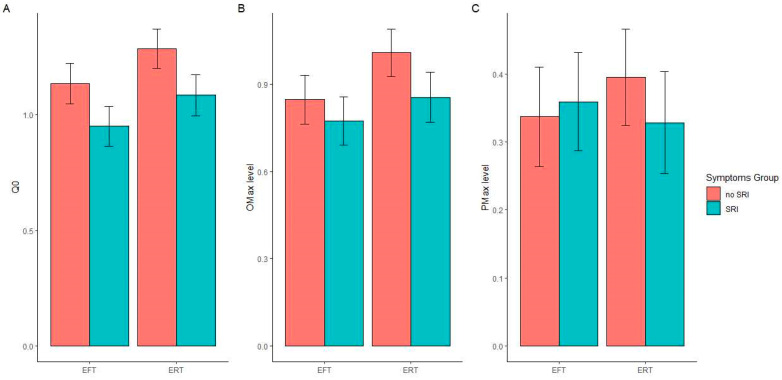
Mean covariate-adjusted log values of Q_0_ (quantity of demand unconstrained by price; (panel (**A**)), Omax (maximum expenditure; panel (**B**)), and PMax (price at which maximum expenditure is observed; panel (**C**)) in the cigarette purchase task in episodic future thinking (EFT) and smoking-related illness (SRI) groups. Higher values of each measure reflect greater demand for cigarettes, with Q_0_ reflecting amplitude of demand, PMax reflecting persistence of demand, and OMax reflecting both amplitude and persistence. Error bars represent standard error of the mean. A significant main effect of SRI was observed on Q_0_ (*p* < 0.050). No other main effects or interactions were significant (in all cases, *p* > 0.100).

### 3.5. Affect

Figure 3 depicts the effects of EFT and SRI on affect scores. No significant main effect of EFT was observed on affect (*F*(1, 194) = 3.584, *p* = 0.060; η_p_^2^ = 0.018), although we did observe a significant main effect of SRI (*F*(1, 194) = 79.189, *p* < 0.001; η_p_^2^ = 0.290), with the latter worsening affect. As in prior analyses, we did not observe a significant EFT × SRI interaction (*F*(1, 194) = 1.914, *p* = 0.168; η_p_^2^ = 0.010). Finally, the covariate effect of income was significant (*F*(1, 194) = 5.289, *p* = 0.023; η_p_^2^ = 0.027), with higher affect scores associated with higher income.

## 4. Discussion

In the present study, EFT significantly reduced delay discounting and cigarette craving. In contrast, SRI symptoms did not significantly reduce but instead *increased* delay discounting compared to the no SRI conditions. However, SRI symptoms did significantly reduce both cigarette craving and demand. Importantly, although both EFT and SRI were able to reduce cigarette craving, only SRI significantly reduced cigarette demand. This suggests that SRI is more effective than EFT in reducing the motivation to smoke.

### 4.1. Combined Effects of Episodic Future Thinking and Smoking-Related Illness

The absence of significant interactions between EFT and SRI conditions in all analyses suggests that the EFT and the EFT + SRI conditions exerted approximately equivalent effects on delay discounting and cigarette craving measures. Effect sizes for interaction terms in these analyses were in the very small range (η_p_^2^ = 0.002–0.005), suggesting the absence of significant interactions was not due to insufficient statistical power. Thus, EFT works to reduce delay discounting and cigarette craving regardless of the presence or absence of SRI symptoms. However, the main effects of SRI symptoms (independent of EFT condition) were more complex, and they demonstrate a dissociation of the typically positive association between delay discounting and measures of cigarette valuation. Specifically, although the effects of SRI symptoms produced therapeutic effects on cigarette craving and demand, SRI increased delay discounting. Thus, although measures of delay discounting and drug valuation often respond to the same experimental variable in similar directions [15,16,18], these measures diverge under some circumstances.

This divergence should perhaps not be surprising, given that cigarette smoking and other drug use are complex phenomena with multiple sources of both environmental and neurobiological control. Accordingly, we note that pre-experiencing SRI symptoms in the present study likely induced stress—a finding supported by SRI’s effects on affect in the PANAS, a measure which correlates strongly with stress exposure [30]. Interestingly, prior data indicate that both acute and chronic stress increases rates of delay discounting [40].Thus, the effects of stress or anxiety may have been responsible for increased delay discounting in the SRI conditions. Another possibility is that imagining a future with lung cancer and its associated mortality produces a greater valuation of immediate over delayed rewards. However, we note that the absence of a significant EFT × SRI interaction on discounting rates does not directly support this latter hypothesis, that is, imagining SRI symptoms produced approximately equal effects on delay discounting regardless of whether these symptoms were in the future or in the recent past. Nonetheless, future work should systematically explore these and other possible mechanisms underlying the effects of SRI on delay discounting.

### 4.2. Concordance with Prior Findings on Effects of Positive Episodic Future Thinking

The present study replicates numerous prior findings on the effects of EFT on delay discounting [13,15,16,17], and it extends these findings to EFT with smoking-related illness. The present study’s effects on cigarette craving also complement those from a prior study [16], demonstrating that EFT reduces laboratory-based cigarette smoking. However, additional findings from the present study differ from recent literature. Specifically, one study [18] reported that EFT significantly reduced cigarette demand (Q_0_) but did not reduce cigarette craving. In contrast, significant and null effects of EFT on demand and craving measures in the present study were reversed. Here, we note that the directional (but nonsignificant) differences in both sets of null findings in these studies were consistent with the therapeutic effects of EFT (e.g., see Figure 2A; Q_0_ EFT effect size, η_p_^2^ = 0.014); thus, perhaps effects of EFT on these measures of smoking value require larger sample sizes to observe robust findings, with unexamined differences in participant characteristics between studies exerting a moderating influence. Future studies should recruit larger samples and should be designed to examine potential moderators of EFT’s effects on measures of smoking value (e.g., readiness to quit, socioeconomic status).

### 4.3. Concordance with Prior Findings on Effects of Negative Episodic Future Thinking

The present study is one of the first to examine the effects of EFT involving emotionally negative content. Only four prior studies to our knowledge have examined emotionally negative EFT—two in which EFT featuring negative content (non-specific to illness) reduced delay discounting [41,42], and two in which negative EFT *increased* discounting [42,43] compared to control conditions.

The reasons for these mixed findings are as yet unclear, and they require additional study. However, we note that none of these prior studies compared negative EFT to a negative control condition; rather, these studies compared EFT to either a neutral episodic-thinking condition without a temporal component [40] or an effectively neutral baseline or no-episodic thinking condition [42,43,44]. If negative emotional valence increases delay discounting, as the present and prior data suggest [39], then a control condition featuring negative emotional content would be required.

### 4.4. Clinical Implications

Both EFT and SRI reduced measures of motivation to smoke, following a relatively brief intervention (approximately 10 min to complete the cue generation tasks). This suggests that EFT methods may be adapted for use as a low-cost intervention strategy for smoking cessation. Prior studies in overweight/obese populations have adapted EFT for use in dietary and weight control [20,45] by engaging participants in EFT repeatedly—but briefly—in the natural environment via smartphone or other web-based methods. The present study’s effects of both EFT and SRI symptoms on smoking behavior suggest that adapting these methods for use in smoking cessation would be similarly worthwhile. Such interventions may be used in isolation or, perhaps more effectively, combined with other evidence-based treatments such as cognitive-behavioral therapy. However, we note that further work is required, particularly in understanding the effects of SRI on delay discounting. Even if SRI symptoms reduce smoking motivation, caution would be warranted with their use because the present findings suggest that SRI may increase other impulsive behavior through its effects on delay discounting. Moreover, the potentially aversive properties of the SRI conditions may reduce either acceptance of or adherence to such an SRI-based treatment.

## 5. Limitations

One limitation of the present study is that our use of AMT yielded a sample that was not representative of the general smoking population (see Table 1). Specifically, we recruited more highly educated and affluent smokers than prior studies (Jamal et al., 2018). Lower socioeconomic status (SES) in particular, has been associated with executive dysfunction, possibly leading to increased impulsivity [4]. Future work should examine the combined effects of EFT and SRI in more representative samples to determine if SES or other demographic variables modulate the findings reported here. Moreover, despite the use of random group assignment, an unexpected and a significant difference in income was observed between groups assigned to SRI conditions and to no-SRI conditions. Although the main effects of these variables on delay discounting and other outcomes were significant when including income as a covariate, future research should consider using stratified block randomization or other methods to ensure groups are balanced on relevant sociodemographic criteria. A second limitation of the present study was the inability to verify the smoking status of AMT participants via physiologic markers (e.g., cotinine), as the study was conducted online. The use of laboratory-based data collection may have corroborated and increased the accuracy of self-report. Future work should explore these effects in a laboratory setting and include such markers to verify smoking status.

## 6. Conclusions

Both traditional EFT and EFT specific to SRI reduced delay discounting compared to their respective EFT control conditions. However, SRI appears to increase delay discounting, independent of the presence or the absence of EFT. Regardless of these divergent effects on discounting, both EFT and SRI conditions reduced cigarette craving, with the SRI condition also reducing behavioral economic demand for cigarettes. Future work should further examine possible mechanisms underlying these effects and the possible utility of both EFT and SRI conditions for use in smoking cessation interventions. Likewise, future work should examine the generality of the effects of EFT specific to the lifestyle-related disease in other populations, such as those suffering from alcohol-use disorders or obesity. Moreover, future work should examine potential moderators of EFT’s findings to elucidate the reasons for heterogeneous findings.

## Figures and Tables

**Figure 3 ijerph-19-07136-f003:**
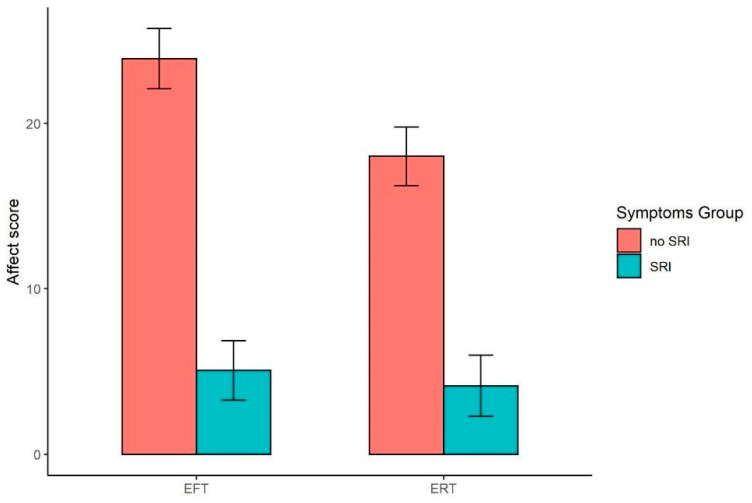
Mean covariate-adjusted affect scores (positive minus negative affect) in episodic future thinking (EFT) and smoking-related illness (SRI) groups. Higher values reflect greater positive vs. negative affect. Error bars represent standard error of the mean. A significant main effect of SRI was observed (*p* < 0.001). No other main effects or interactions were significant (in all cases, *p* > 0.050).

**Table 2 ijerph-19-07136-t002:** Numbers of participants whose delay discounting and cigarette demand data were identified as nonsystematic by individual criteria.

		Group			
Measure	Criterion	EFT	EFT-SRI	ERT	ERT-SRI
Delay discounting	Trend	15	5	7	4
	Bounce	0	0	2	0
Cigarette demand	Trend ^a^	2	8	0	7
	Bounce	0	0	0	0
	Reversal	0	0	0	0

^a^ All violations of the trend criterion were due to zero purchasing at all prices (i.e., null demand).

## Data Availability

A publicly available dataset was analyzed in this study. This dataset can be found here: https://osf.io/fqt8z/?view_only=25baf105bb154694b740d010b3a7d030 (accessed on 29 April 2022).

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
