# Peer review of "Episodic Future Thinking about Smoking-Related Illness: A Preliminary Investigation of Effects on Delay Discounting, Cigarette Craving, and Cigarette Demand"

_ijerph, 2022, doi:10.3390/ijerph19127136_

Round 1

Reviewer 1 Report

The authors have used an online sample to examine the effects of EFT involving SRI on delay discounting, and two different measures of motivation to smoke including cigarette craving, and behavioral economic demand for cigarettes. They also compared these effects to EFT without SRI. They used a factorial design using a 2 (episodic thinking) x 2 (SRI). The findings suggested a significant effect on delay discounting for EFT (reduced discounting) and SRI (increased discounting), and of EFT and SRI on cigarette craving where cigarette craving was reduced. Overall, they suggested that these methods might have clinical implication to help with smoking cessations interventions.

The authors have done a nice work of examining the research question and have presented a lot of details in terms of background and analysis. A few minor comments:

  • Towards the end of the introduction, it seems like there is too much detail presented for the objectives. It will help if the objectives were shortened, and the authors should consider if lines 94-98 are required to explain the objectives.
  • How was the sample size determined? Was a power analysis done?
  • What was the criteria used to determine to be a current smoker? The authors mention smoking ≥ 5 cigarettes but how was this determined?
  • The methods have been described in great detail, but it seems like a lot of information. Would it be possible to reduce some of the text maybe provide additional details in a supplementary section?

Author Response

We thank Reviewer 1 for their comments. Below, we outline how we have responded in this revised manuscript.

  • Towards the end of the introduction, it seems like there is too much detail presented for the objectives. It will help if the objectives were shortened, and the authors should consider if lines 94-98 are required to explain the objectives.
    • We thank the reviewer for this suggestion. We have shortened this section of the introduction by moving the material in line 94-98 to the Method section.
  • How was the sample size determined? Was a power analysis done?
    • In the revised manuscript , we now specify that sample size was chosen to match prior EFT studies. We now also present results of a sensitivity power analysis, in which our approximate sample size yielded 95% power to detect a medium effect size or larger in ANOVA.
  • What was the criteria used to determine to be a current smoker? The authors mention smoking ≥ 5 cigarettes but how was this determined?
    • In the revised manuscript, we now clarify that participants completed a screening questionnaire prior to the study.
  • The methods have been described in great detail, but it seems like a lot of information. Would it be possible to reduce some of the text maybe provide additional details in a supplementary section?
    • We thank the reviewer for this comment. We recognize that the methods section is detailed. However, we believe this level of detail is necessary to readers’ understanding of the methods and analysis, given the novel procedures used in the present study. Although we considered shortening this section, the revised manuscript leaves the description of methods in tact.

Author Response

We thank Reviewer 2 for their comments. Below, we outline how we have responded in this revised manuscript.

  • If there are references to criteria for effect size, please indicate.
    • We have added reference to Cohen (1988).
  • In Table 1, the frequency (%) notation and the mean (SD) notation are mixed, so it seems necessary to distinguish them. Also consider expressing the frequency as n(%) and the mean as Mean±SD.
    • In the revised table, we have clarified the reporting of frequency and mean values.
  • Could differences in income have anything to do with racial ratios?
    • We appreciated this comment, as it led us to revisit demographics of the four groups. Ultimately, however, there is no evidence that the group differences in income were meaningfully related to race differences due to: 1) the very low frequency of non-white race categories, and 2) race did not differ significantly between groups. These analyses are reported in the original and revised manuscripts; thus, no additional revisions have been made.
  • When writing two figures, consider how to mark them separately as (a) and (b). Please indicate the value on the graph.
    • In the revised figures, the different panels are noted alphabetically.
  • Check again how to write 95% confidence intervals.
    • Confidence intervals have now been reported appropriately throughout.
  • Because statistical results tables are not presented in the study results, it is difficult for the reader to read and interpret the exact values. Exact values are also not given in the figure. Rather than reading the text to understand statistical results, it seems to increase readability by presenting it as a table.
    • We prefer to leave the figures unrevised, although we now present full ANCOVA results in a supplementary table.
  • Please indicate the same number of decimal places with 3 or 2 digits.
    • In the revised manuscript, we now report all p values to 3 decimals throughout.
  • When a researcher hypothesized about an interaction effect, there must have been considerable evidence. Nevertheless, no interactions were found in the study results. Do you have any suggestions for further research?
    • We appreciate the reviewer’s attention to this matter. However, we had no specific hypotheses about interactions in this preliminary and exploratory study. This is now clarified at the end of the Introduction.